# Natural Oil-Based Rigid Polyurethane Foam Thermal Insulation Applicable at Cryogenic Temperatures

**DOI:** 10.3390/polym13244276

**Published:** 2021-12-07

**Authors:** Katarzyna Uram, Aleksander Prociak, Laima Vevere, Ralfs Pomilovskis, Ugis Cabulis, Mikelis Kirpluks

**Affiliations:** 1Department of Chemistry and Technology of Polymers, Faculty of Chemical Engineering and Technology, Cracow University of Technology, Warszawska 24, 31-155 Cracow, Poland; aleksander.prociak@pk.edu.pl; 2Polymer Laboratory, Latvian State Institute of Wood Chemistry, Dzerbenes Street 27, LV-1006 Riga, Latvia; laima.vevere@kki.lv (L.V.); ralfs.pomilovskis@kki.lv (R.P.); ugis.cabulis@kki.lv (U.C.); mikelis.kirpluks@kki.lv (M.K.); 3Faculty of Materials Science and Applied Chemistry, Institute of Technology of Organic Chemistry, Riga Technical University, P. Valdena St. 3/7, LV-1048 Riga, Latvia

**Keywords:** bio-polyols, rigid polyurethane foams, cryogenic insulation

## Abstract

This paper presents research into the preparation of rigid polyurethane foams with bio-polyols from rapeseed and tall oil. Rigid polyurethane foams were designed with a cryogenic insulation application for aerospace in mind. The polyurethane systems containing non-renewable diethylene glycol (DEG) were modified by replacing it with rapeseed oil-based low functional polyol (LF), obtained by a two-step reaction of epoxidation and oxirane ring opening with 1-hexanol. It was observed that as the proportion of the LF polyol in the polyurethane system increased, so too did the apparent density of the foam material. An increase in the value of the thermal conductivity coefficient was associated with an increase in the value of apparent density. Mechanical tests showed that the rigid polyurethane foam had higher compressive strength at cryogenic temperatures compared with the values obtained at room temperature. The adhesion test indicated that the foams subjected to cryo-shock obtained similar values of adhesion strength to the materials that were not subjected to this test. The results obtained were higher than 0.1 MPa, which is a favourable value for foam materials in low-temperature applications.

## 1. Introduction

Conventional polyurethane (PUR) materials are obtained through a polyaddition reaction between polyisocyanate and polyol [1]. The PUR market was valued at over $45 billion by 2020, and 65% of their production are porous materials [2]. This market is increasing every year due to growing consumer industries such as furniture and bedding, automotive, electronics and construction. However, up to 65% of the market is focused on foam materials that contain an open- or closed-cell structure. Flexible foams have an open-cell structure, and their most popular application is in the furniture industry. Closed-cell rigid PUR foams are mainly used as thermal insulation materials. Thanks to their low thermal conductivity value, they contribute to reducing heat loss, promoting energy-efficient construction. These good thermal insulation properties result from the closed-cell structure of the rigid PUR foam [3].

The majority of PURs feedstock is derived from the petrochemical origin, despite societies striving towards more sustainable solutions [4,5]. The growing awareness and concern for sustainability require the exploitation of natural resources. A beneficial solution to link the PUR industry with sustainable development is the production of polyols of natural origin. Bio-polyols can be used as substrates for solid (adhesives, films) and porous materials (flexible, rigid foams) [6]. Replacing petrochemical raw materials with renewable ones increases the proportion of biomass in the materials and makes rigid PUR foams more environmentally friendly [7].

The number of publications concerning the preparation of bio-polyols from natural oil for rigid PUR foams is increasing every year. Oils are esters of glycerol and higher saturated and unsaturated fatty acids. The absence of hydroxyl groups in the most popular vegetable oils (except castor oil) necessitates a chemical reaction to introduce reactivity with isocyanates, which are another component of PUR material production. Various oils such as rapeseed, soybean, palm, coconut, sunflower and castor oils have been used for bio-polyol development. Their applicability depends on the geographical region where the availability of different oils is variable [8,9,10]. The advantage of using vegetable oils as feedstock for bio-polyols is low price, easy availability and renewable [11]. In addition, life cycle assessments of vegetable oil-based polyols indicate reduced consumption of petroleum-based feedstocks [12].

There are several methods described in the literature for modifying oils and introducing hydroxyl groups: hydroformylation, ozonolysis and hydrogenation, epoxidation and opening oxirane rings. The mentioned methods are mainly concerned with the modification of unsaturated bonds occurring in fatty acid residues. Another way of modification is the processes of transesterification and transamidation, in which ester groups are modified to hydroxyls [13]. The chemical structure of bio-polyols from vegetable oils differs from that of polyols derived from non-renewable raw materials. The differences in chemical structure between the petrochemical polyol and bio-polyol are related to the use of different starting raw materials and the methods of obtaining hydroxyl derivatives.

The properties of rigid PUR foams are affected by numerous factors. For example, the functionality and hydroxyl value of the bio-polyol significantly affect the cross-linking density, which in turn affects the cell structure formed during the foaming process. Ugarte et al. [14] prepared rigid PUR foams using blends of a high-functional polyol derived from sorbitol and a diol derived from corn sucrose. They found that the cross-link density of the produced rigid PUR foams is directly modified by the polyol blend ratio. A higher corn-derived diol content in the PUR system reduces the cross-link density. The cross-linking density depends on the number of urethane bonds formed during the foaming process and, thus, on the number of hydroxyl groups present in the polyol. The presence of dangling hydrocarbon chains derived from higher fatty acid residues has a significant effect on the properties of final foams. Vegetable oil-based polyols cause a plasticising effect that reduces the mechanical strength of the resulting materials [8,15,16,17]. Furthermore, the urethane bond content determines the properties of the resulting foam materials. Their higher content of urethane bond in PUR foam increases the stiffness of the material and, thus, results in higher compressive strength values. In addition, the effect of increased compressive strength is also a result of a higher presence of hydrogen bonds formed between the urethane-urethane, urethane-ester and urethane-ether groups [18].

Rigid PUR foam can operate efficiently over a wide temperature range (−235 to 150 °C) due to its low thermal conductivity values. This material has good mechanical strength and insulation, energy absorption and has corrosion resistance. Therefore, it is one of the most widely used materials in cryogenic systems [19]. The literature indicates that these materials can successfully insulate tanks and pipelines delivering liquefied natural gas (LNG) [20]. Another application is to insulate the external tanks of the space shuttle, which contain liquid oxygen and hydrogen that are commonly used as rocket fuel. Rigid PUR foams have been used in the space industry for many years due to their properties, i.e., low thermal conductivity (k = 0.02 W/m·K at room temperature) and high strength-to-density ratio (8–9 kPa/kg/m^3^) [21]. The main advantage of rigid PUR foams is the possibility to cover this material on the complicated shape metal surfaces by a spraying method. The properties of the material and its adhesion to the substrate depend not only on the chemical structure and macromolecular architecture of the polymer matrix but also on the technological factors of PUR foam production [22].

This paper presents the possibility of obtaining rigid PUR foams based on bio-polyols from renewable raw materials as thermal insulation materials, which can be used at cryogenic temperatures. The tailoring of the PUR polymer matrix properties was performed by changing the ratio of different polyols in the foam formulation. A balance between soft and hard segments, as well as the addition of dangling chains into the PUR polymer matrix, was used to develop a rigid PUR foam material that can be applied as cryogenic insulation. Rigid PUR foams need to be relatively flexible at cryogenic temperatures but also rigid at room temperature for cryogenic insulation application. Bio-polyols were obtained from rapeseed and tall oils using different synthesis methods. The aim of this study was to replace diethylene glycol (DEG), which is a petrochemical material, with low functional (LF) polyol obtained by a two-step reaction: epoxidation followed by an oxirane ring opening reaction. The effects of LF polyol on the foaming process, cell structure and mechanical properties of the obtained materials were analysed. In addition, the behaviour of the material before and after cryo-shock was studied.

## 2. Materials and Methods

Rigid PUR foams were obtained using bio-polyols and petrochemical components. Two types of bio-polyols (HF polyol and LF polyol) were prepared at Cracow University of Technology, as described in previous works [23,24]. Bio-polyols were synthesised in a two-step process using rapeseed oil as the raw material. In the first step, unsaturated bonds in the oil were oxidised by peroxyacid, produced in-situ between acetic acid and hydrogen peroxide. The resulting oxirane rings were then reacted with two different alcohols (1-hexanol and 1,6-hexanediol) to generate hydroxyl groups. Bio-based cross-linking polyol based on epoxidised tall oil fatty acids (ETOFA) and triethanolamine (TEOA) was synthesised (ETOFA/TEOA polyol), as described in previous work [25]. Bio-polyol was prepared by the Latvian State Institute of Wood Chemistry. Commercially available polyol NEO 380 (Neopolyol 380) is an aromatic polyester polyol based on industrial polyethylene terephthalate (PET) waste and purchased from NEO Group (Klaipeda, Lithuania). The properties of the polyol raw materials used are shown in Table 1. DEG was used as the chain extender polyol, and it was supplied by Sigma Aldrich (St. Louis, MO, USA). Diphenylmethane diisocyanate (pMDI) with an isocyanate group content of 31.5% was supplied by Covestro (Leverkusen, Germany). The formulation used a catalytic system containing amine and organometallic catalysts to accelerate gelation and foaming reactions. Surfactant Niax silicone L-6915 was delivered from Momentive Performance Materials (Waterford, NY, USA). Two different types of blowing agents were used. Distilled water was used as a chemical blowing agent, and a mixture of hydrofluorocarbons was used as a physical blowing agent. In addition, tris(2-chloroisopropyl) phosphate (TCPP) purchased from Albermarle (Belgium) was used as a flame retardant to protect the rigid PU foams from fire hazards.

### 2.1. Preparation of Rigid PUR Foams

Rigid PUR foams were prepared using the free-rise method. At the first stage, a polyol premix, consisting of a mixture of polyol components, catalysts, blowing agents, surfactant and flame retardant, was prepared. The components were mixed for 60 s until a homogeneous mixture was obtained. A sufficient amount of isocyanate was then introduced to give an isocyanate index of 110. The polyurethane system was mixed for 10 s and poured into the vertical open-top mould with dimensions of 200 mm × 200 mm. Obtained rigid PUR foams were conditioned at room temperature for 24 h to complete the cross-linking process. The formulations of the prepared foams are shown in Table 2.

### 2.2. Characteristics of Rigid PUR Foams

Rigid PUR foams modified by bio-polyols were tested to determine their physical, thermal and mechanical properties. The foaming process was analysed using a FOAMAT device (Format-Messtechnik GmbH, Karlsruhe, Germany). The temperature during the process was measured with a thermocouple. Foam shrinkage was measured after 24 h of the foaming process using an ultrasonic sensor. The morphology of cells was analysed using an optical microscope (PZO, Warsaw, Poland). The anisotropy index was calculated as the ratio of the cell height and width. Cell density was determined according to equation [26]:(1)$$NF={\left(\frac{n}{A}\right)}^{\frac{3}{2}}$$
where: *n* is s the number of bubbles in the micrograph of area *A* in cm^2^.

The closed-cell content was measured according to the ISO 4590:2016 standard [27]. Apparent density was calculated as a ratio of masses and volumes of samples according to the ISO 845:1995 standard [28]. Thermal conductivity was determined using a Linseis Heat flow meter 200, and the test was carried out according to ISO 8301:1991 [29]. The sample of 200 mm × 200 mm × 50 mm was placed between two plates (top plate temperature 20 °C, bottom plate temperature 0 °C).

Compressive strength in 10% of deformation was measured using Zwick Z010 TN (Zwick GmbH & Co, Ulm, Germany) for foams at a temperature of 293 K. The tests were performed in two directions—parallel (Z) and perpendicular (X) to foam growth—and cylinders of 20 mm diameter and 20 mm height were used as samples. For the compression test in cryogenic (77 K) temperatures, similar samples as for testing at 293 K were used together with a testing jig that can be submerged into a liquid nitrogen container. The tensile properties were determined by testing two different shape samples. For tensile tests at a temperature of 293 K, dumbbell-shaped samples (length 153 mm, thickness 12 mm, width 440 mm and bottleneck 26 mm) were used, whereas for tensile tests at a temperature of 77 K, ring-type samples were used (width 13 mm, inner diameter 43 mm and outer diameter 53 mm). The tensile measurement was performed only in a parallel direction using Zwick Z010 TN (Zwick GmbH & Co, Ulm, Germany) due to a rigid PUR foam sample block height limitation. The different mechanical properties of the testing jigs are depicted in Figure 1. To test compression and tensile properties at cryogenic temperatures, the samples, together with the testing jig, were submerged into liquid nitrogen. It can be easily performed for the small cylindrical-shaped samples for the compression strength test. Unfortunately, it was not feasible to submerge into liquid nitrogen the relatively large dumbbell-shaped samples together with the cumbersome clamps. Thus, a different sample size was used, i.e., rigid PUR foam rings, which were pulled apart by steel half-cylinders (Figure 1.) For all tests at liquid nitrogen temperatures, the sample was conditioned in liquid nitrogen for 5 min before testing. Detailed analysis of this method has been described in [30]. For each rigid PUR foam formulation, a minimum of five replicates of the mechanical tests were performed. Moreover, for the tensile strength tests at a temperature of 77 K, eight replicate samples were tested.

Adhesion of the cryogenic insulation PUR foams to aluminium was measured as tensile strength before and after cryo-shock (immersion in liquid nitrogen for 60 min, then warming to room temperature). Samples were tested according to PN-EN 1607:2013 [31] at a temperature of 293 K. Rigid PUR foam specimens of 40 mm × 40 mm × 20 mm were used, which were glued to aluminium plates of the same dimensions using an epoxy-based adhesive.

## 3. Results and Discussion

### 3.1. Foaming Process

The first stage of the work was to investigate the foaming process of tested PUR compositions, which shapes the cell structure of the final porous materials. Many factors affect the formation of the cellular structure, including the type of raw materials used, the growth rate of the material and the temperature inside the foam core, which are influenced by catalyst system and hydroxyl components reactivity [32]. Parameters of foaming processes measured using the FOAMAT device are shown in Table 3.

Analysing the characteristic foaming times of PUR systems, it was observed that for all PUR formulations, the start time was in the range of 25 to 30 s. However, the core temperature during the foaming process decreased with an increase in the LF polyol content. This effect is beneficial when rigid PUR foams are used as thermal insulation materials because it prevents the foam core from overheating, which may lead to foam failure due to thermal degradation. A similar relationship was observed for gel time, in which the time did not exceed 100 s.

Replacement of DEG with LF polyol showed that the apparent densities of rigid PUR foams (obtained during the foaming in the FOAMAT device) increased. The increase in apparent density of the materials is related to a decrease in foaming of the polyol premix. Moreover, the LF polyol has a higher viscosity than DEG, which affects cell formation and leads to problems during the PUR foaming process. Song et al. observed that excessive viscosity can lead to inhibition of foam bubble formation and growth [33]. In other studies, it has been observed that increasing the viscosity of the polyol premix causes difficulty in forming the cell structure [34,35]. Another reason for the increase in apparent density is the volume of gas formed during the foaming process. Lower temperatures of reaction mixtures during the foaming process mean that the gas phase takes up a smaller volume. This phenomenon results from Clapeyron’s equation, which states that the volume of gas depends on its temperature. Despite the increasing apparent density of rigid PUR foams, these materials were dimensionally stable (Table 3).

### 3.2. Cell Structure and Physical Properties of Developed Rigid PUR Foams

The course of the foaming process has a significant effect on the formation of the cell structure of rigid PUR foams. Small changes in the PUR system affect the cellular structure, which is responsible for the thermal insulation properties of these materials. Parameters of the cell structure of obtained foams are shown in Table 4, and selected cell structure images are shown in Table 5.

The anisotropy coefficient of cells in the foams was defined as the ratio of cell height to cell width. A value of 1 for this parameter means that the foam cells are spherical in shape. The anisotropy coefficient values for PP40-PP44 foams indicated differences in the shape of the cells analysed in the cross-section parallel and perpendicular to the foam growth direction. Cells in the cross-section parallel to the foam growth direction reach an anisotropy coefficient above 1 and are characterised by an elongated shape [36]. In contrast, cells in the perpendicular section have a factor of less than 1. Differences can also be observed in the values of foam cell cross-sectional area and cell density. The cells in the parallel cross-section are significantly larger compared with the cells in the cross-section perpendicular to the foam growth direction. Moreover, the cross-sectional areas of the foam cells were related to the cell density, which clearly indicated that fewer cells with larger cross-sectional areas were obtained. Replacement of DEG with LF polyol did not influence the differences in values of anisotropy index, a cross-sectional area of foams and cell density for PP40-43 systems. Nevertheless, disordered cell structure was clearly visible in the system designated as PP44. Despite the similar shape, cells with a much larger cross-sectional area with a simultaneous decrease in cell density were observed. The change of rigid PU foam morphology is explained by the increased viscosity of the LF polyol.

The formation of the cell structure of foams during the foaming process has a significant impact on the properties of the resulting materials [37]. In the case of rigid PUR foams, which are one of the most popular thermal insulation materials, the shape, surface and type of cells are of significant importance. Selected physical properties (apparent density, thermal conductivity and closed-cell content) of the foams are presented in Table 6.

The apparent density of the rigid PUR foams (Table 6) obtained in vertical moulds differed from the values obtained during FOAMAT measurements (Table 3). Nevertheless, the trend of change in apparent density was similar in both cases. The lower values of apparent density of rigid PUR foams obtained in the mould were due to the larger volume of the foamed material. The expansion of the foamed material is not construed as in the case of the cup tests in the FOAMAT equipment.

Thermal conductivity values of the compared foams correlate with the values of closed-cell content and apparent density and are similar to what is described in the literature [38]. Higher content of closed cells resulted in the materials with lower thermal conductivity coefficients and, thus, better thermal insulating properties. With the increase in LF polyol content in the PUR system, the value of thermal conductivity of the foams also increased due to the effect of decreasing closed-cell content. The reduction in the closed-cell content allowed faster diffusion of the physical blowing agent from the rigid PU foam. Moreover, increasing thermal conductivity values were also associated with an increasing apparent density as the thermal conductivity through the polymer matrix increased. The foam designated as PP44 had the highest thermal conductivity value when compared with the other foams. The reason for this was the disruption of the cell structure during the foaming process, which caused the cells of the rigid PUR foam to open up. The effect of increasing the heat transfer coefficient in conjunction with foam cell opening was also observed by Tan et al. [39] and Członka et al. [40].

### 3.3. Mechanical Properties

Obtained data from compression experiments were normalised with respect to the apparent density of 50 kg/m^3^. The equation used for normalisation of modulus (*E_norm_*) and strength (*σ_norm_*) are the following [41]:(2)$$E_{norm}=E{\left(\frac{50}{\rho_{sample}}\right)}^{1.7}$$
(3)$$\sigma_{norm}=\sigma {\left(\frac{50}{\rho_{sample}}\right)}^{2.1}$$

The compressive modulus and strength were measured in both parallel and perpendicular directions at two different temperatures: 293 K (room temperature (RT)) and 77 K (cryogenic temperature (CT)). Results of compressive modulus and strength are shown in Figure 2.

The mechanical strength of rigid PUR foams depends on many factors, including the type of raw materials used, the apparent density of the material, and its cellular structure, which is formed during the foaming process [42]. The results clearly indicate that the values of compressive strength and compressive modulus for rigid PUR foams were lower at RT than CT. For CT temperatures, the compressive strength values ranged from 0.25 to 0.50 MPa, while for RT temperatures, the values were about 0.1 MPa. The higher values obtained for mechanical tests at cryogenic temperatures were due to the higher stiffness of rigid PUR foams at lower temperatures. Moreover, in both cases, it was observed that the values obtained for the tested systems in the direction parallel to the direction of foam growth were higher than in the perpendicular direction. The differences in the values in different directions were due to the anisotropic structure of the foams, which were confirmed by the anisotropy coefficient (Table 4). It is worth noting that the parameter tested under different temperature conditions indicates that these materials have a good compressive strength of over 0.1 MPa. In the case of the rigid PUR foams with the highest content of LF polyol, the highest difference in compressive strength and modulus were noticed when comparing the results in parallel and perpendicular directions to the foam growth direction. This foam also had the highest anisotropy coefficient and average cell area in both foam cross-sections. Increasing the LF polyol content in the PUR system resulted in a decrease in compressive strength and compressive modulus at both RT and CT. Replacement of short DEG chains with an LF polyol containing long, dangling chains derived from hydrocarbon residues of higher fatty acids resulted in a plasticising effect. A similar effect was observed in other studies when bio-polyols that contain dangling chains were applied [8,43].

Tensile modulus (Young’s), strength and elongation at break were also measured at RT and CT. These tests were performed only in the direction parallel to the direction of foam growth, and the results are shown in Figure 3.

Replacement of DEG with LF polyol in the PU system resulted in reduced tensile strength and elongation at break in both temperatures. The changes in Young’s modulus for individual samples were small and did not depend on the amount of LF polyol in the PUR formulation. Similar to the compressive strength tests, higher modulus values were obtained at cryogenic temperatures. For example, the values of Young’s modulus for RT were 8–10 MPa, while for CT, they were much higher, varying from 17 to 22 MPa. The changes in tensile strength indicated that this property much more depends on LF polyol content in foam formulation when the measurement is carried out at CT. When tested at CT, higher tensile strength values were obtained for PP40 and PP41 samples compared with those obtained at RT. The similar values of tensile strength at CT and RT for the samples containing more than 5 pbw of LF polyol are due to the plasticising effect of the polyurethane matrix, causing a decrease in material stiffness. The higher tensile strength values for these samples were due to both the higher stiffness of the foam materials at low temperatures and the content of hard segments formed from urethane bonds. The higher content of DEG, the higher amount of isocyanate was used, increasing the content of hard segments. In addition, hydrogen bonds present in the structure of rigid PUR foam also affect the strength values. The effect of the increased internal strength of materials resulting from the presence of hydrogen bonds in hard segments was observed in the work of Stirna et al. [18] and Xu et al. [44]. The unusually high standard deviations of the mechanical properties determined at cryogenic temperatures may be explained by the relatively small sample size. A small defect in the rigid PUR foam structure may play a significant role in the deformation behaviour of the material, whereas for the elongation at break data at RT, the high standard deviations are explained by the nature of the developed rigid PUR foam material. At RT conditions, the foam is somewhat viscoelastic, and it slipped out of the clamps used for the tensile strength test. Despite the relatively high standard deviations, it is possible to observe the trend of LF polyol influence the properties of the developed bio-based rigid PUR foams.

An opposite effect was obtained for the elongation tests, in which the values at room temperature were higher than at cryogenic temperature. Rigid PUR foam based on bio-polyols had a higher elasticity at room temperature. Increasing the amount of LF polyol in the PUR system resulted in a decrease in the elongation values. This change is related to the disruption of the cell structure of the rigid PUR foam. The smallest values were obtained for system PP44, where the cell structure was disrupted. The elongation values for this sample at RT and CT temperatures were 4% and 2%, respectively.

The safety coefficient (see Table 7) is a characteristic of the materials and describes its ability to maintain adhesion to the aluminium plate after cryo-shock. It is calculated by the following equation:(4)$$k_{S}=\frac{\varepsilon_{77}}{\Delta l_{77-300}}=\frac{\varepsilon_{77}}{\alpha_{x}*\Delta T*100}$$

*ε*_77_—tensile elongation at 77 K, %

Δ*l*_77–300_—shrinkage of material cooling it from 300 to 77 K, %

*α_x_*—coefficient of thermal expansion, 10^−6^/°C

Δ*T*—temperature gradient, degrees

For cryogenic insulation applications, rigid PUR foam usually is applied on different metal surfaces, such as steel or aluminium tanks. One of the main characteristics of this type of material is its ability to stick to the surface of the substrate, i.e., adhesion strength. The advantage of rigid PUR foams is their good adhesion to many surfaces [45], including metals. Furthermore, rigid PUR foam maintains its low thermal conductivity at low temperatures, down to 20 K [46].

The rapid cooling of the material can cause the introduction of microcracks in the rigid PUR material, which can cause lower adhesion strength and even delamination for the substrate surface. This is tested by so-called cryo-shock when the samples are immersed in liquid nitrogen, and their adhesion properties are tested afterwards.

Increasing the LF polyol content caused the adhesion to the aluminium plate to decrease (decreasing safety coefficient, Table 7). The systems containing 10 or more parts of LF polyol had a constant value of this coefficient. It can be assumed that the introduction of more LF polyol will not affect the ability of the rigid foam to adhere to aluminium plates. An adhesion of obtained rigid PUR foam to the metal surface and its behaviour after immersion in liquid nitrogen were tested, and the results are shown in Figure 4.

The results showed that the introduction of LF polyol into the PUR system resulted in a decrease in the adhesion strength values compared with the system without this bio-polyol. The results obtained are higher than 0.1 MPa, which for foam materials is a favourable value at low-temperature applications. Moreover, the results clearly indicated that the cryo-shocked samples obtained similar values of adhesion modulus strength and elongation at break. These materials can be successfully used as insulation in space, where the temperature is close to the temperature of liquid nitrogen.

## 4. Conclusions

The results presented in this paper indicated that the obtained rigid PUR foams can be successfully used as thermal insulation materials at low temperatures. Replacing DEG with LF polyol with low functionality did not change the characteristic foaming times but led to lower foaming process temperatures. Moreover, the cell structure of the rigid PUR foams was disturbed due to the higher viscosity of the LF polyol. It was also observed that the value of the thermal conductivity coefficient increased as a consequence of the lower content of closed cells of rigid PUR foam. The mechanical tests indicated that more favourable mechanical strength values were obtained at cryogenic temperatures. Rigid PUR foams have shown increased stiffness at low temperatures. The presented studies clearly indicate that rigid PUR foams can be successfully used as thermal insulation materials at cryogenic temperatures.

## Figures and Tables

**Figure 1 polymers-13-04276-f001:**
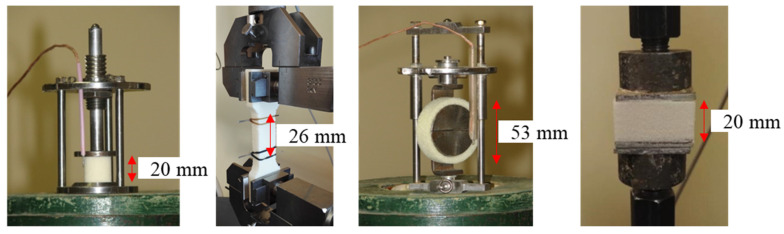
Sample of rigid PUR foam for compression tests and tensile tests at 293 K, tensile tests at 77 K and with aluminium plates prepared for adhesion test.

**Figure 2 polymers-13-04276-f002:**
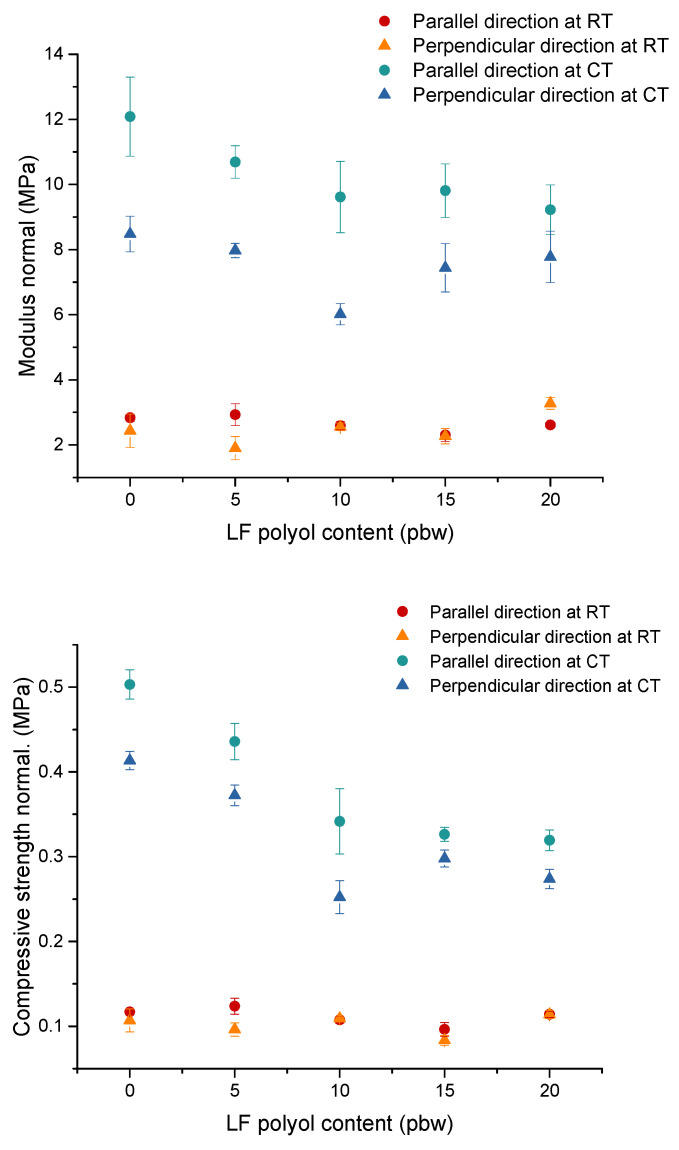
Compressive modulus and strength of foams at RT and CT.

**Figure 3 polymers-13-04276-f003:**
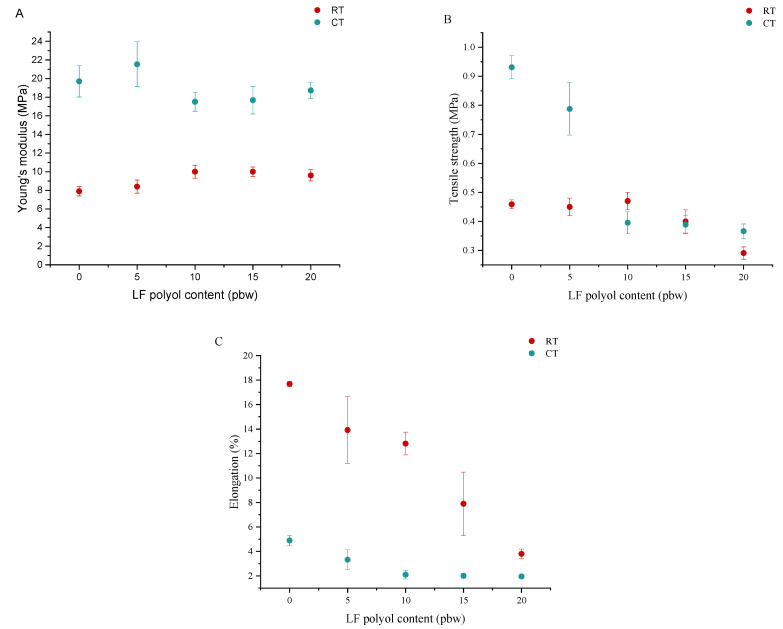
Tensile modulus (Young’s) (**A**), tensile strength (**B**) and elongation at break (**C**) of foams at RT and CT.

**Figure 4 polymers-13-04276-f004:**
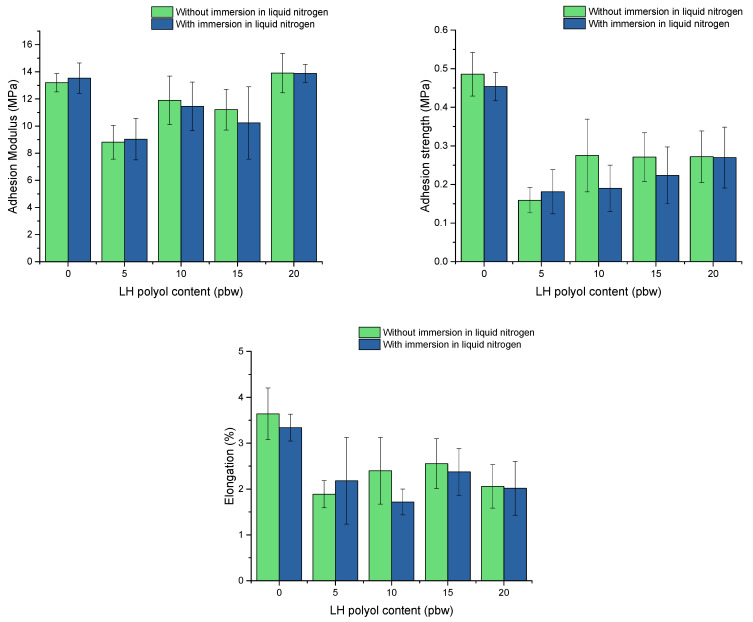
Adhesion modulus, strength and elongation at break of rigid PUR foams without and after immersion in liquid nitrogen.

**Table 1 polymers-13-04276-t001:** Characteristic of different polyols used for rigid PUR foam development.

Name of Polyol	HF Polyol	LF Polyol	ETOFA/TEOA	NEO 380	DEG
OH_val_ mgKOH/g	267	115	448	370	1057
Functionality	4.7	3.0	7.1	3.3	2.0
Water content, %	0.40	0.05	0.14	0.21	0.17

**Table 2 polymers-13-04276-t002:** Formulations of rigid PUR foams.

Component	PUR Formulation Name				
	PP40	PP41	PP42	PP43	PP44
ETOFA/TEOA	25	25	25	25	25
NEO 380	20	20	20	20	20
HF polyol	30	30	30	30	30
**LF polyol**	**0**	**5**	**10**	**15**	**20**
**DEG**	**25**	**20**	**15**	**10**	**5**
Flame retardant	15	15	15	15	15
Physical blowing agent	25	25	25	25	25
Chemical blowing agent	0.5	0.5	0.5	0.5	0.5
Catalysts	0.5	0.5	1.0	1.0	1.5
Surfactant	1.5	1.5	1.5	1.5	1.5
pMDI	147	147	144	145	142
**Sustainable material content in PU foam, %**	**16.1**	**17.3**	**18.7**	**19.9**	**21.3**

**Table 3 polymers-13-04276-t003:** Parameters of foaming processes of PUR systems PP40-PP44 (data from FOAMAT).

PUR Foam System	t_start_, s	t_rise_, s	T_max_, °C	Apparent Density, kg/m^3^	Shrinkage, %
PP40	26.3 ± 0.4	82.7 ± 0.2	154.6	52.7 ± 1.1	0.2 ± 0.2
PP41	30.2 ± 0.4	102.1 ± 0.2	151.7	55.9 ± 0.9	0.0 ± 0.1
PP42	31.3 ± 0.2	92.8 ± 0.2	143.1	60.3 ± 1.6	0.1 ± 0.2
PP43	27.0 ± 0.2	89.3 ± 0.2	133.9	63.1 ± 0.9	0.1 ± 0.1
PP44	25.2 ± 0.2	84.5 ± 0.2	137.3	61.6 ± 0.2	0.1 ± 0.1

**Table 4 polymers-13-04276-t004:** Selected cell structure parameters of rigid PUR foams.

PUR Foam System	Direction of Growth	Anisotropy Coefficient	Cross-Section Area, 10^3^ mm^2^	Cell Density, (Number of Cells 10^3^/cm^3^)
PP40	Parallel	1.22 ± 0.04	12.4 ± 1.8	307 ± 52
	Perpendicular	0.90 ± 0.03	8.6 ± 0.2	868 ± 58
PP41	Parallel	1.27 ± 0.02	12.8 ± 2.7	304 ± 73
	Perpendicular	0.80 ± 0.01	8.0 ± 1.2	825± 14
PP42	Parallel	1.39 ± 0.02	16.0 ± 4.5	257 ± 103
	Perpendicular	1.00 ± 0.03	9.8 ± 1.4	714 ± 160
PP43	Parallel	1.18 ± 0.08	12.7 ± 5.9	261 ± 14
	Perpendicular	0.82 ± 0.06	8.8 ± 1.6	837 ± 366
PP44	Parallel	1.18 ± 0.08	27.5 ± 2.7	24 ± 3
	Perpendicular	0.80 ± 0.06	29.9 ± 1.1	46 ± 34

**Table 5 polymers-13-04276-t005:** Optical microscope images of rigid polyurethane foams PP40-44.

**Cross-Section type**	**Rigid PUR Foam Formulation**				
	**PP40**	**PP41**	**PP42**	**PP43**	**PP44**
**Parallel Direction**	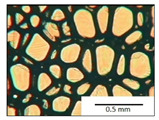	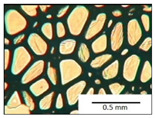	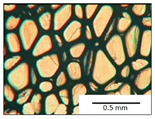	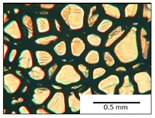	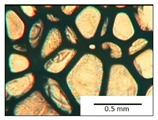
**Perpendicular Direction**	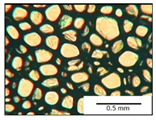	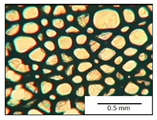	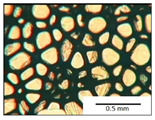	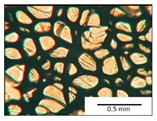	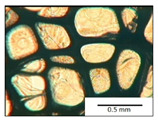

**Table 6 polymers-13-04276-t006:** Selected physical properties of rigid PUR foams.

PU Foam System	Apparent Density, kg/m^3^	Thermal Conductivity, mW/(m∙K)	Closed-Cell Content, %
PP40	49.7	18.72	97.5 ± 0.4
PP41	49.9	18.85	92.0 ± 1.0
PP42	52.2	20.09	91.0 ± 1.0
PP43	53.2	21.35	88.0 ± 2.0
PP44	49.4	27.51	78.0 ± 1.0

**Table 7 polymers-13-04276-t007:** Linear thermal expansion and safety coefficient of developed rigid PUR foams.

PU Foam System	LF Polyol Content (pbw)	α (10^−6^/°C)	Safety Coefficient
PP40	0	85.39	2.5
PP41	5	85.66	1.7
PP42	10	78.30	1.1
PP43	15	72.19	1.1
PP44	20	75.09	1.0

## Data Availability

The data presented in this study are available on request from the corresponding author.

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
