# Peer review of "Natural Oil-Based Rigid Polyurethane Foam Thermal Insulation Applicable at Cryogenic Temperatures"

_polymers, 2021, doi:10.3390/polym13244276_

Round 1

Reviewer 1 Report

The authors present here fabrication and characterization of polyurethane foam based on renewable biopolyols for thermal insulation applications. It is interesting topic for development of natural oil-based rigid polyurethane foam by changing the formulation for efficient insulation management. However, this manuscript lacks some vital information. Also, the discussions for this version to support the big achievement of these fields are weak. Therefore, authors need the revision of the manuscript for publishing the work for publication in Polymers journal. Some questions and suggestions are followed;

[1] We recommend that authors change the name of PUR foam system and PUR formulation such as PUR40, 41, 42, 43, and 44 in order to increase identification of samples by journal readers. 

[2] We suggest that authors should add the scale bar into Figure 1 and Table 5, respectively.

[3] We suggest that authors should increase font size in Figure 3 and Figure 4 in order to increase understanding of this manuscript by journal readers.  

[4] The form of references described in References part does not match with the guideline of the Polymers journal. The authors should revise references’ form accurately.

Author Response

Dear Reviewer,

Thank you for your review and quick reply. I am grateful for all your comments, which are very valuable and will enrich the manuscript. Changes included in revised manuscript have been marked with red colour.

Yours sincerely,

Review 1

The authors present here fabrication and characterization of polyurethane foam based on renewable biopolyols for thermal insulation applications. It is interesting topic for development of natural oil-based rigid polyurethane foam by changing the formulation for efficient insulation management. However, this manuscript lacks some vital information. Also, the discussions for this version to support the big achievement of these fields are weak. Therefore, authors need the revision of the manuscript for publishing the work for publication in Polymers journal. Some questions and suggestions are followed;

  1. We recommend that authors change the name of PUR foam system and PUR formulation such as PUR40, 41, 42, 43, and 44 in order to increase identification of samples by journal readers.

Thank you for your recommendation. However, we do not want to change the name of the PUR foam system and PUR formulation because the results published in the article is only part of the research results. Other results, where the best PUR foam formulation PP40 is filled with different fillers, is at the final stages of being accepted for publishing in Polymers and references this paper. Due to this reason, we want to keep the naming of samples unified. For better identification of the samples, Figures 2, 3, and 4 show the changes in polyol LH concentration.

  1. We suggest that authors should add the scale bar into Figure 1 and Table 5, respectively.

Thank you for the kind suggestion. Scale bars for Figure 1 and Table 5 have been added.

  1. We suggest that authors should increase font size in Figure 3 and Figure 4 in order to increase understanding of this manuscript by journal readers.

This part has been completed.

  1. The form of references described in References part does not match with the guideline of the Polymers journal. The authors should revise references’ form accurately.

This part has been corrected.

Reviewer 2 Report

Article is interesting in my opinion. However, some things must be improved before it could be published in Polymers.

1) All standards mentioned in Subsection 2.2 should be listed in Reference list. Now, readers must search what is the scope of each standard.

2) Authors mentioned about few kinds of the samples in Subsection 2.2. Scheme with the geometry of each sample should be presented in new Figure to avoid misunderstanding.

3) Line 168: "Detailed analysis of this method has been described in [26]". Most important information about this method should be added into revised version of the manuscript.

4) How the mechanical tests (compression and tensile) was performed in cryogenic temperature ? There is a lack of this information in the text.

5) Figure 2 - why the compressive strength in case of the pbw10 have such big standard deviation in compare to the other tested samples ?

6) Figure 3 - why the standard deviation in case of pbw5(CT) sample is so huge ? It should be explained in text.

7) Figure 3 - when polyol content is higher than 5, then the tensile strength of the samples is no longer higher in CT than in RT. Why ? It should be also explained in text.

8) How many samples of each type was used during the research ?

9) Authors prepared few foam systems (PP40-PP44). For which foam system results of the mechanical tests are presented ? Why authors do not present results for all prepared foam systems ?

10) Authors should be more specific in description of the obtained results. Results should be compared with use of numerical values (not only words like "higer", "smaller"). Changes in trends in the charts (Figure 2,3) should be also explained in text.

Author Response

Dear Reviewer,

Thank you for your review. I am grateful for all your comments, which are very valuable and will enrich the manuscript. Changes included in revised manuscript have been marked with red colour.

Yours sincerely,

Article is interesting in my opinion. However, some things must be improved before it could be published in Polymers.

  1. All standards mentioned in Subsection 2.2 should be listed in Reference list. Now, readers must search what is the scope of each standard.

This part has been added.

  1. Authors mentioned about few kinds of the samples in Subsection 2.2. Scheme with the geometry of each sample should be presented in new Figure to avoid misunderstanding.

This part has been completed.

  1. Line 168: "Detailed analysis of this method has been described in [26]". Most important information about this method should be added into revised version of the manuscript.

This part has been completed.

  1. How the mechanical tests (compression and tensile) was performed in cryogenic temperature? There is a lack of this information in the text.

Tests in cryogenic temperature were performed using a similar procedure as at 293 K. The Zwick apparatus was equipped with a tank containing liquid nitrogen. Samples were placed in special holders before testing, as shown in Figure 1 and then conditioned in liquid nitrogen for 5 minutes. After this time, the following measurements were carried out.

A description of this measurement has been added.

  1. Figure 2 - why the compressive strength in case of the pbw10 have such big standard deviation in compare to the other tested samples ?

A higher than usual standard deviation of the mechanical properties obtained at cryogenic temperatures could be due to the small size of the tested samples (dimensions of about 20 mm). Furthermore, the tests at cryogenic temperatures could have higher operator error as the samples must be placed into a relatively complicated jig. This must be done using cumbersome gloves as the procedure is done at liquid nitrogen temperature.

  1. Figure 3 - why the standard deviation in case of pbw5(CT) sample is so huge ? It should be explained in text.

Thank you for the keen observation. The explanation for the large standard deviation of the tested samples is similar to the previous comment. In the case of the tests done at cryogenic temperatures, the deviation between parallel samples is due to small sample dimensions as the imperfections of the foamed material play a much more significant role on the mechanical deformation of the sample. In case of the elongation at break for the samples tested at room temperature there is another problem. We used dumbbell shape PUR foam samples. Developed foams at room temperature are relatively flexible, as the material has been developed with cryogenic insulation in mind. Some of the tested samples slipped out from the perforated pneumatic jaws. Thus, the high standard deviation for the elongation at break. Furthermore, the developed rigid PUR foams could not be ideally homogeneous. This is not seen from the appearance of the foam materials. Recently (i.e. yesterday) we have noticed that there is an unusually fast separation of the polyol components. This was not noticed before as the foams were prepared relatively soon after polyol premix preparation. If the polyol components separate fast and make an unstable emulsion this could lead to domains of PUR polymer matrix with drastically different mechanical properties. This is something that we are investigating further and the PUR foam formulations will be optimized.

Nevertheless, we believe that despite the high standard deviation, it is possible to see the trend of LF polyol influence on the properties of developed bio-based rigid PUR foams.

  1. Figure 3 - when polyol content is higher than 5, then the tensile strength of the samples is no longer higher in CT than in RT. Why ? It should be also explained in text.

This part has been added. (The similar values of tensile strength at CT and RT for the samples containing more than 5 pbw of LF polyol are due to the plasticizing effect of the polyurethane matrix, causing a decrease in material stiffness.)

  1. How many samples of each type was used during the research ?

When testing rigid polyurethane foams, a minimum of 5 replicates were performed for each system. This number of specimens is sufficient to understand the behaviour of the specimen in a given test. For the tensile tests at liquid nitrogen temperatures, a higher amount - 8 of parallel samples were tested.

  1. Authors prepared few foam systems (PP40-PP44). For which foam system results of the mechanical tests are presented ? Why authors do not present results for all prepared foam systems ?

This paper presents 5 different polyurethane systems in which diethylene glycol was replaced by LF polyol (Table 2).  All rigid polyurethane foams were analyzed and the results of mechanical tests are shown in Figures 2 and 3.

  1. Authors should be more specific in description of the obtained results. Results should be compared with use of numerical values (not only words like "higer", "smaller"). Changes in trends in the charts (Figure 2,3) should be also explained in text.

Thank you for the comment, the text has been altered accordingly.

Round 2

Reviewer 2 Report

Article is much better now. However, Authors used various kind of samples during tensile test - it should be explained why they do that and how it affected into results(see Point 1). Some explanations from the previous round should be added into revised version of the manuscript (see Point 2).

1) In case of tensile test performed in 293 K dumbbell shape samples were used (”dumbbell shape samples (length 153 mm, thickness 12 mm, width 440 mm and bottleneck 26 mm) were used for tensile tests in temperature 293 K”). However, in 77K Authors used other kind of the samples (“For tensile tests, ring type samples were used”). In next line Authors wrote “Tests in cryogenic temperature were performed in the same procedure as at 293 K”. In my opinion it could be confusing for the readers because Authors used different kind of samples during the tests. Why Authors used various kind of samples during tensile test? How did this affect the results? It should be explained in the text.

2) Explanations from Remarks 5-6, 8 (previous round) should be added to the manuscript.

Author Response

Dear Reviewer,

Thank you kindly for the review. We have improved the article accordingly and the revised changes are depicted in the manuscript in purple colour.

Yours sincerely,

Article is much better now. However, Authors used various kind of samples during tensile test - it should be explained why they do that and how it affected into results (see Point 1). Some explanations from the previous round should be added into revised version of the manuscript (see Point 2).

1) In case of tensile test performed in 293 K dumbbell shape samples were used (”dumbbell shape samples (length 153 mm, thickness 12 mm, width 440 mm and bottleneck 26 mm) were used for tensile tests in temperature 293 K”). However, in 77K Authors used other kind of the samples (“For tensile tests, ring type samples were used”). In next line Authors wrote “Tests in cryogenic temperature were performed in the same procedure as at 293 K”. In my opinion it could be confusing for the readers because Authors used different kind of samples during the tests. Why Authors used various kind of samples during tensile test? How did this affect the results? It should be explained in the text.

For compression test in cryogenic (77 K) temperatures similar samples as for testing at 293 K were used together with a testing jig that can be submerged into a liquid nitrogen container. The tensile properties were determined by testing two different shape samples. For tensile tests in temperature 293 K a dumbbell shape samples (length 153 mm, thickness 12 mm, width 440 mm and bottleneck 26 mm) were used. Whereas for tensile tests in temperature 77 K ring-type samples were used (width 13 mm, inner diameter 43 mm and outer diameter 53 mm). The tensile measurement was performed only in a parallel direction using Zwick Z010 TN (Zwick GmbH & Co) due to a rigid PUR foam sample block height limitation. The different mechanical properties testing jigs are depicted in Figure 1. To test compression and tensile properties at cryogenic temperatures, the samples together with the testing jig was submerged into liquid nitrogen. It can be easily done for the small cylindrical shape samples for the compression strength test. Unfortunately, it is not feasible to submerge into liquid nitrogen the relatively large dumbbell shape samples together with the cumbersome clamps. Thus, different sample size was used, rigid PUR foam rings, that were pulled apart by steel half-cylinders (see Fig. 1.) For all tests at liquid nitrogen temperatures, the sample was conditioned in liquid nitrogen for 5 min before testing. Detailed analysis of this method has been described in [30]. For each rigid PUR foam formulation a minimum of 5 replicates of the mechanical tests were performed. Moreover, for the tensile strength tests in temperature 77 K 8 replicate samples were tested.

2) Explanations from Remarks 5-6, 8 (previous round) should be added to the manuscript.

The manuscript text was revised accordingly (see page 10).

An unusually high standard deviations of the mechanical properties determined at cryo-genic temperatures may be explained by the relatively small sample size. A small defect in the rigid PUR foam structure may play a significant role in the defamation behaviour of the material. Whereas for the elongation at break data at RT the high standard deviations are explained by the nature of the developed rigid PUR foam material. At RT conditions the foam is somewhat viscoelastic and it slipped out of the clamps used for the tensile strength test. Despite the relatively high standard deviations it is possible to see the trend of FL polyol influence on the properties of the developed bio-based rigid PUR foams.
